# Identification of Allergens in White- and Red-Fleshed Pitaya (*Selenicereus undatus* and *Selenicereus costaricensis*) Seeds Using Bottom-Up Proteomics Coupled with Immunoinformatics

**DOI:** 10.3390/nu14091962

**Published:** 2022-05-07

**Authors:** Mengzhen Hao, Ziyi Zhao, Huilian Che

**Affiliations:** College of Food Science and Nutritional Engineering, China Agriculture University, Beijing 100083, China; mengzhen.hao@cau.edu.cn (M.H.); xijir0304@163.com (X.); ziyiiz@163.com (Z.Z.)

**Keywords:** pitaya seeds, *Selenicereus undatus*, *Selenicereus costaricensis*, food allergen, proteomics, IgE-binding proteins

## Abstract

White-fleshed pitaya (*Selenicereus undatus*) and red-fleshed pitaya (*Selenicereus costaricensis*) are becoming increasingly popular because of their nutritional and medicinal benefits. However, in addition to their beneficial properties, allergy to pitaya fruits has occurred in daily life. In this study, we investigated the protein profile of pitaya fruit seeds and focused on the most reactive proteins against immunoglobulin E (IgE) in sera from allergic patients by immunoblotting. A protein band of approximately 20 kDa displayed a clear reaction with the serum IgE. The protein bands of interest were excised, in-gel digested, and analyzed using liquid chromatography–tandem mass spectrometry (LC–MS/MS), followed by data searching against a restricted database (*Caryophyllales* in UniProtKB) for protein identification. Immunoinformatic tools were used to predict protein allergenicity. The potential allergens included cupin_1 and heat shock protein 70 (HSP70) in white-fleshed pitaya seeds, and cupin_1, heat shock protein 70, and heat shock protein sti1-like in red-fleshed pitaya seeds are potential allergens. The expression of potential allergens was further verified at the transcriptional level in the species of *S. undatus* and *S. costaricensis.*

## 1. Introduction

Pitaya fruit is commonly known as dragon fruit and belongs to the genus *Selenicereus* in the family *Cactaceae*, and its cultivation has increased because of reports highlighting promising medicinal uses of these plants [1,2]. The edible seeds of pitaya fruit resemble seeds of kiwi fruit and are embedded in the pulp [3], accounting for 8% of the total weight of pitaya fruit [4]. Pitaya seed oil contains a high number of lipids and can be used as a source of essential oils, which contain fatty acids, sterol, phenols, and tocopherol [5]. The major phytoconstituents isolated from pitaya seed oil have health-promoting properties, such as antioxidant activity [6]. Although the medicinal and nutritional value of pitaya seeds have been widely reported, the potential allergenicity risk of seeds in the edible part of pitaya fruits should not be ignored because the seeds represent a main source of protein. The proteins that occur in the seeds of edible fruits might behave as hidden allergens and trigger hypersensitive responses wrongly attributed to pulp allergens [7]. Recently, we found cases of suspected allergy to white- and/or red-fleshed pitaya, and the allergic symptoms after the consumption of this fruit included vomiting, urticaria, etc. Pitaya seeds were the prime suspect causing IgE hypersensitivity to pitaya fruits.

Preliminary investigations have reported the allergenic potential of pitaya fruits and indicated that they are capable of inducing immune reactions in sensitive individuals. Using serum from a 30-year-old woman in Spain, the 15 kDa protein detected via IgE immunoblotting represents an allergen in red-fleshed pitaya, and it is speculated to be profilin according to the patient sensitized to profilins, as previously demonstrated by positive skin prick tests [8]. In the following year, a serological test revealed that a lipid transfer protein (LTP) of approximately 10 kDa in red-fleshed pitaya is an allergen, using serum from a 19-year-old woman from Germany [9]. These studies have demonstrated that there are IgE-binding proteins in the whole pitaya fruits. However, whether allergens are hidden in the seeds of white- and red-fleshed pitayas requires further investigation.

In the last decade, liquid chromatography (LC) coupled with mass spectrometry (MS) has played an indispensable role in the identification and characterization of food allergens. Particularly free of protein separation is bottom-up proteomics combined with the in-gel technique, which is based on protein digestion to generate peptides and LC–MS/MS to identify peptides in large quantities [10]. This method is used to screen many unexpected potential allergens. LC–MS/MS can be used for protein identification in large quantities, but it is also prone to produce redundant results because the peptides alignments depend on protein databases. With the identification of an increasing number of allergens and their structures, immunoinformatic tools in silico have been developed to evaluate the potential allergenicity of proteins and analyze epitopes based on protein homology and three-dimensional structure [11]. Therefore, to eliminate the redundant results of LC–MS/MS for the identification of potential allergens, immunoinformatic tools in silico can be added to the workflow.

In the present study, we investigated immunoreactive proteins to IgE from white- and red-fleshed pitaya seeds and performed a proteomic investigation of the most reactive proteins from pitaya fruit seed extracts as observer-based, immunoblot experiments performed on the sera of five patients sensitized to pitaya fruits. Our results confirmed that one band at approximately 20 kDa was the most relevant for immunoreactivity, and the individual spots were further subjected to MS analysis and protein identification. In combination with immunoinformatic tools analyzing proteins allergenicity online, two and three potential allergens in white- and red-fleshed pitaya seeds were identified, respectively. The corresponding gene transcription was detected to determine the protein expression and gene sequence in white- and red-fleshed pitaya seeds.

## 2. Materials and Methods

### 2.1. Patient Sera Collection

Sera were collected from five patients, and the collection and use of sera were reviewed and approved by the Human Research Ethics Committee of China Agricultural University (CAU-HR2021015). The patients stated that they have allergic symptoms after eating white- and/or red-fleshed pitaya. Blood samples were centrifuged at 3000× *g* and 4 °C for 10 min following coagulation for 1 h at room temperature to obtain sera. The sera were stored at −80 °C before analysis. The blood collection process was approved by the patients. The characteristic information of pitaya allergic patients is shown in Appendix A.

### 2.2. Preparation of Protein Extract

The fresh white- and red-fleshed pitaya samples were purchased from local retailers. Referred to Galvez et al. [12], an extraction protocol optimized in the present study. Briefly, seeds of white- and red-fleshed pitaya were collected in water using degreased gauze. Mature seeds were ground mechanically and the powder was de-oiled with n-hexane at room temperature overnight. The powder was subsequently air-dried and kept at −20 °C until required. Seed proteins were extracted in 0.025 M Tris–0.192 M glycine buffer (pH = 8.3) with a solid–liquid ratio of 1:10 (*w*/*v*) at 4 °C for 12 h. The suspension was centrifuged at 12,000× *g* for 15 min, and the supernatant was filtered through a 0.22 μm membrane and then stored at −20 °C. The protein concentration of the extracts was determined by a BCA kit (Solarbio Inc., Beijing, China) using bovine serum albumin as a standard.

### 2.3. Reduced SDS–PAGE Analysis

Reduced SDS–PAGE was performed using 4~20% polyacrylamide gels (Dakewe Inc., Beijing, China) and a Tris–Gly–SDS running buffer (50 mM Tris, 40 mM glycine, 3 mM SDS). Reduced 16.5% Tris–tricine gels (Solarbio Inc., Beijing, China) and a Tris–tricine–SDS running buffer (100 mM Tris, 100 mM Tricine, 3 mM SDS) were used to separate the lower MW proteins. Pre-stained protein molecular weight standards of 11, 17, 25, 35, 48, 63, 75, 100, 135, 180 kDa (Biorigin Inc., Beijing, China) and low-molecular-weight standards of 1.7, 4.6, 10, 15, 25, 40 kDa (Thermo Fisher Scientific, San José, CA, USA) were used as references. Crude protein samples were boiled in a reduced SDS sample buffer (Biorigin Inc., Beijing, China) for 5 min before loading. Gels were stained in R-250 staining solution (0.1% Coomassie brilliant blue R250, 10% acetic acid, and 45% methyl alcohol).

### 2.4. Immunoblotting with IgE

Western blot was performed according to the method described by Guo et al. [13]. Proteins were subjected to 4~20% polyacrylamide gel electrophoresis (SDS–PAGE), and the separated proteins were transferred to polyvinylidene fluoride (PVDF) membranes (Millipore Inc., Darmstadt, Germany). The membranes were blocked in 0.01 M Tris-buffered saline containing 0.1% Tween 20 (Xilong Science Co., Ltd., Shantou, Guangdong Province, China) and 5% non-fat milk powder (Wako Pure Chemical Inc., Osaka, Japan), and then incubated with sera overnight at 4 °C. Next, the membranes were incubated with a specific peroxidase-conjugated goat anti-human IgE antibody (Sigma-Aldrich Inc., St. Louis, MO, USA) for 2 h at room temperature. Finally, the membranes were incubated in enhanced chemiluminescence (ECL) reagent (Millipore Inc., Darmstadt, Germany) and then exposed and developed. The images were analyzed using a chemiluminescence developer (Clinx Science Instruments Co., Ltd., Shanghai, China).

### 2.5. LC–MS/MS Proteomic Investigation and Immunobioinformatics Analysis

The most informative protein bands of pitaya seeds were cut from 4~20% polyacrylamide gels and submitted to the in-gel digestion by trypsin procedure, according to the protocol reported by De Angelis et al. [14]. Peptide mixture resulting from the digestion of pitaya fruit seeds protein extract was analyzed by liquid chromatography–tandem mass spectrometry (LC–MS/MS) system, followed by a bioinformatic search for identification purposes. Specifically, a Q-Exactive^TM^ Plus Hybrid Quadrupole-Orbitrap^TM^ Mass Spectrometer coupled with a high-performance liquid chromatography (HPLC) pump system (Thermo Fisher Scientific, San José, CA, USA) was used. Samples were loaded through an automatic sampler to a pre-column C18 trap column (5 μm, 300 μm × 5 mm, 100 Å) and then separated using an analytical column C18 column (3 μm, 75 μm × 150 mm, 100 Å), with a flow rate of 300 nL/min. After processing the MS raw spectra, protein identification was achieved by searching against the *Caryophyllales* database (https://www.uniprot.org/uniprot/?query=taxonomy:3524 accessed on 14 December 2021), obtained from UniProtKB. MASCOT software. The database-searching parameters were as follows: enzyme: trypsin; max missed cleavages: 1; peptide mass tolerance: ±20 ppm; fragment mass tolerance: 0.6 Da; peptide confidence: high; the filtration parameters were as follows: peptide FDR ≤ 0.05.

Sequences of the top 10% of proteins identified by LC–MS/MS were input into Allermatch^TM^ (http://allermatch.org/ accessed on 14 December 2021), Algpred 2.0 (https://webs.iiitd.edu.in/raghava/algpred2/ accessed on 14 December 2021) and AllerCatPro (https://allercatpro.bii.a-star.edu.sg/ accessed on 14 December 2021) to predict protein allergenicity. 

### 2.6. RT-PCR and Cloning

Total RNA was prepared from mature pitaya fruit seeds using a TRIzol^TM^ Reagent (Thermo Fisher Scientific Inc., Waltham, MA, USA), according to protocol performed by Donald C et al. [15]. First-stand DNA was synthesized using a cDNA synthesis kit (TransGen Inc., Beijing, China) with an oligo (dT) adaptor. The resulting first-strand cDNA was used as a template for PCR amplification. PCR was performed using HiFi DNA Polymerase (TransGen Inc., Beijing, China). The PCR reaction protocol was as follows: initial denaturation at 95 °C for 4 min, followed by 35 cycles of denaturation at 95 °C for 30 s, annealing at 57 °C for 30 s, and extension at 72 °C for 1 min, with an additional extension at 72 °C for 10 min. The RT-PCR products were analyzed by 1% agarose electrophoresis with a TAE buffer (40 mM Tris, 2 mM EDTA disodium salt dihydrate, 20 mM acetic acid). The primers for RT-PCR were designed in NCBI based on mRNA sequences for potential allergens in European Nucleotide Archive (ENA) database. The primers are outlined in Appendix A.

## 3. Results

### 3.1. Gel Electrophoresis Separation of Pitaya Seed Proteins

The total protein concentrations from 10 g of seeds of white- and red-fleshed pitaya extracted in 100 mL buffer solution were 6.89 ± 0.01 mg/mL and 6.80 ± 0.02 mg/mL. The whole pitaya seeds’ protein extracts were loaded onto SDS gels for protein separation to obtain the final protein profiles under reduced conditions. As shown in Figure 1a, the three most intense protein bands migrated in three main areas of the 4~20% Tris–Gly–SDS polyacrylamide gels and covered the molecular weight (MW) ranging 11~17 kDa, 17~25 kDa, and 75 kDa. In addition, a diffuse band was also observed at the bottom of the gel below the 11 kDa band, which could not be well defined in 4~20% Tris-Gly–SDS polyacrylamide gels.

Furthermore, 16.5% Tris–tricine gels were used to separate proteins in the lower MW range close to 10 kDa. Although this further separation confirmed the presence of certain proteins at the higher MW, which were already clearly visualized in the 4–20% gel, the proteins in the medium- and low-MW range appeared to be better resolved in the 16.5% Tris-tricine gels. As shown in Figure 1 b, two bands were detected in the range of 15~25 kDa, corresponding to the 4~20% Tris–Gly–SDS polyacrylamide gels. Below 10 kDa, two bands of approximately 9 and 7 kDa in the 6.5% Tris–tricine gels.

### 3.2. Immunoblot Analysis for IgE Binding

The proteins separated on the 4~20% Tris–Gly–SDS and 16.5% Tris–tricine electrophoresis gel were electroblotted on a 0.22 μm PVDF membrane before incubation with each individual serum of allergic patients. The images acquired from an immunoblot for IgE developed after incubation with the serum of the five allergic patients are shown in Figure 2 and Figure 3.

The most reactive white-fleshed seed proteins were concentrated in the area of 20 kDa, which is highlighted with arrows in the figures. No bands reacted against IgE in the lower MW, which was further confirmed using immunoblot analysis after protein separation on 16.5% Tris–tricine polyacrylamide gels. In addition, the IgE reactivity profiles on 4~20% Tris–Gly–SDS polyacrylamide gels appeared to be very similar to those on 16.5% Tris–tricine polyacrylamide gels.

The distribution of IgE binding to red-fleshed pitaya seed proteins on 4~20% Tris–Gly–SDS polyacrylamide gels was similar to that of white-fleshed pitaya seed proteins, which was concentrated at20 kDa. At approximately 40 kDa, proteins were also bound to IgE from several sera.

As shown in Figure 2 and Figure 3, the protein bands at approximately 20 kDa from both white- and red-fleshed pitaya showed reactivity to IgE in most of the patients’ sera. Therefore, the 20 kDa bands that proved to react against allergic patients’ sera were cut from the 4~20% Tris–Gly–SDS polyacrylamide gels and subsequently prepared for LC–MS/MS analysis for final protein identification.

### 3.3. LC–MS/MS Identification of the IgE-Binding Protein Bands

The IgE-binding protein bands of approximately 20 kDa in the 4~20% Tris–Gly–SDS polyacrylamide gels were excised with label “a” shown in Figure 1. The staining protein bands were decolored and submitted to in-gel trypsin digestion before LC–MS/MS analysis. All collected data were further processed using the commercial software MASCOT, which enabled straightforward identification of the candidate proteins by interrogating large protein databases available online (UniProtKB). The *Caryophyllales* database was used for protein screening. In the gel spots labeled “a” from white- and red-fleshed pitaya seed protein extractions, 53 and 178 proteins were identified, respectively. The total proteins retrieved using the MASCOT discoverer software screening are listed in Appendix A. As shown in Table 1 and Table 2, the top 10% of proteins are listed by score. And the distributions of identified peptides in the top 10% of protein identified via LC–MS/MS in ex-cised gel spot from white- and red-fleshed pitaya seeds were shown on Appendix A, respectively.

In the gel spots labeled “a”, proteins identified from white-fleshed pitaya seeds were mostly found from red-fleshed pitaya seeds (4/5). These consistent proteins from two different samples were identified as cupin type-1 domain-containing protein (A0A7C8ZNT3), cupin-1 (A0A0K9RTZ6), oil-body-associated proteins (OBAPs) (A0A803LTG6), and heat shock protein 70 (HSP 70) (A0A1I9TK81).

### 3.4. Immunoinformatic Analysis for Identified Proteins Using Allergen Prediction Platform Online

To further narrow down the range of potential allergens, Allermatch^TM^, Algpred 2.0, and AllerCatPro were used to predict the allergenicity of proteins identified by LC–MS/MS. The results for proteins from white- and red-fleshed pitaya seeds predicted as allergens on all three online platforms are shown in Table 3 and Table 4, respectively. Protein families to which potential allergens from pitaya seeds and corresponding similar allergens belonged are listed in these tables. Classifications of protein families were searched on InterPro or Pfam. The allergen name and the routes of exposure to corresponding known allergens were accessed from WHO/IUIS ALLERGEN NOMENCLATURE (http://www.allergen.org/ accessed on 14 December 2021), ALLERGOME (http://www.allergome.org/ accessed on 14 December 2021), or COMPARE (http://db.comparedatabase.org/ accessed on 14 December 2021) databases. All details on protein allergenicity predicted by the three online platforms are described in Appendix A.

Of four consistent proteins identified by LC–MS/MS in the two different pitaya seed samples, three proteins (A0A7C8ZNT3, A0A0K9RTZ6, and A0A1I9TK81) are predicted as allergens. The predicted allergens from two different pitaya seeds belonged to the cupin_1 and heat shock protein 70 (HSP70) protein families. Moreover, in the red-fleshed pitaya seeds, two other proteins (A0A7C9B091 and I0CC94) are also predicted as allergens and classified as heat shock protein Sti1-like and AhpC-TSA protein families. Using Allermatch^TM^ and AllerCatPro platforms, the most similar predicted allergens were output. It was found that protein families to which potential allergens in white-fleshed and red-fleshed pitaya seeds belong are similar to the corresponding known allergens, which are mostly classified as vicilin-like proteins, heat shock proteins, and AhpC-type proteins. The routes of similar allergen exposure have been recorded on the WHO/IUIS ALLERGEN NOMENCLATURE and ALLERGOME databases.

### 3.5. RT-PCR Analysis for Expression of Potential Allergen in White- and Red-Fleshed Pitaya Seeds on the Level of mRNA

The predicted allergen sources identified using LC–MS/MS were *opuntia Streptacantha*, *Spinacia oleracea*, and *Tamarix hispida.* These predicted allergens have not been recorded in the protein databases of white-fleshed pitaya (*Selenicereus undatus)* and red-fleshed pitaya (*Selenicereus costaricensis*), whether the identified proteins were expressed in these two species or these proteins identified by LC–MS/MS only for similar enzyme-hydrolyzed peptides occurred in the pitaya species and the above three species needed to be verified.

mRNA sequences of the predicted allergens were obtained from the European Nucleotide Archive (ENA) database. The mRNA sequences were BLAST-searched against the *S. undatus* genome (assembly ASM1758966v1 in the NCBI genome database) [16] for determining whether genes encoding predicted allergens were presented. The BLAST results are shown in Table 5. Except for the protein accessed A0A0K9RTZ6 and I0CC94 in UniPortKB, the mRNA sequences of the other three proteins successfully aligned with the *S. undatus* genome. These results were further verified by RT-PCR, as shown in Figure 4. Based on the BLAST and RT-PCR results, the coding sequences (CDSs) for amino acids in allergenic proteins in white- and red-fleshed pitaya were inferred, as shown in Appendix A. *Su-cupin1* and *su-hsp70* are important genes encoding potential allergens in white-fleshed pitaya. In comparison, *su-cupin1*, *su-hsp70*, and *sc- hsp sti 1* are important genes encoding potential allergens in red-fleshed pitaya seeds.

## 4. Discussion

White- and red-fleshed pitaya are popular fruits for their good taste and potential pharmacotherapeutic benefits [17,18]. The seeds are hidden in the flesh of pitaya fruits, which cannot be avoided during eating. However, these seeds contain many kinds of proteins that may be dangerous to certain sensitive people, because of their potential allergenicity, such as Act d 12 and Act d 13 in green kiwi fruit seeds [19]. Allergic symptoms have been observed in sensitive individuals after eating fleshed pitaya fruits, and we speculated that the seeds of white- and red-fleshed pitaya represented the chief causative agents of these reactions. Therefore, we studied the protein profiles of fleshed pitaya seeds and focused on the immunoreactive protein bands in the sera of allergic patients.

The protein profile of the white-fleshed pitaya seeds was similar to that of the red-fleshed pitaya seeds in 4~20% Tris–Gly–SDS polyacrylamide gel. Small-molecular-weight proteins were better separated on 16.5% Tris–tricine–SDS polyacrylamide gels. The 9 kDa and 7 kDa bands may be assigned to two subunits of 2S albumin, which have molecular-weight characteristics similar to those of other 2S albumin found in seeds, such as Ses i 1 [20] and Ses i 2 in sesame seeds [21]. Moreover, the 9 kDa band may have represented LTP because it was similar to the LTP in other seeds, such as Ara h 9 in peanuts [22]. There were differences in protein profiles on 16.5% Tris–tricine polyacrylamide gels between the two pitaya seeds. In the white-fleshed pitaya seeds, some proteins were distributed on approximately 11 kDa bands, but this was not observed in red-fleshed pitaya seeds.

Proteins with a molecular weight of approximately 20 kDa readily bound to specific IgE from allergic patients’ sera. LC–MS/MS proteomics indicated that the number of proteins in the 20 kDa gel bands from red-fleshed pitaya seeds was greater than that from white-fleshed pitaya seeds, which indicated that red-fleshed pitaya had greater protein abundance. The top 10% of proteins identified using LC–MS/MS mostly belonged to *O. streptacantha* and *S. oleracea* but not to *S. undatus* or *S. costaricensis.* This was because proteins from *S. undatus* and *S. costaricensis* in the UniProtKB database are few, i.e., 118 and 7, respectively; therefore, the peptides obtained from in-gel trypsin digestion could not be matched with the proteins in the database of *S. undatus* and *S. costaricensis*. Thus, the *Caryophyllales* database in UniProtKB was used to match peptide cleavage by trypsin for more protein identification.

Most allergens are found in a limited number of protein families [23]. Proteins on IgE-binding bands from pitaya seeds were classified into families to which most known allergens belong. Allergenic members of the cupin superfamily belong to seed storage globulins. These proteins are major food allergens from legumes, nuts, and seeds, and include Act d 12 in kiwifruit seeds [19], Coc n 1 in coconut [24], etc. Allergens belonging to the HSP70 family are found in a heterogeneous range of sources, including animals, plants, and fungi. Several animal allergens belong to this family, including Tyr p 28 from storage mites [25]. In addition, the hazelnut pollen allergen Cor a 10 from hazelnut [26] belongs to this family, and the fungal allergen Mala s 10 is also HSP70 [27].

The proteomic results provided a considerable amount of information about the proteins. However, the bottom-up proteomics of peptides matching depends on matching databases and may produce redundant results. Therefore, to eliminate some redundant results, proteomic analysis is generally followed by evaluation methods in silico to predict the potential allergenicity of proteins. Some proteins from novel nutrient sources with high protein contents were evaluated for allergenicity using the strategy presented above, such as proteins extracted from *Spirulina* and *Chlorella* microalgae [28]. Bianco et al. assessed sequence similarities between microalgal proteins identified via LC–MS/MS and known allergens in the AllergenOnline allergen database based on criteria established by the World Health Organization and Food and Agriculture Organization, and this method is the principle underlying the Allgermatch^TM^ allergenicity prediction tool online [29]. Here, the Algpred 2.0 and AllerCatPro alignment methods were performed in silico, in addition to the Allermatch^TM^. On Algpred 2.0, ensemble approaches have been used to predict allergenic proteins by combining prediction scores of different approaches, including BLAST search, IgE epitopes alignment, motif elicitation/motif alignment, and machine learning techniques [30]. The AllerCatPro prediction web server considers conformational epitopes with similar local structures [31]. Three web servers were comminated for predicting protein allergenicity from different features of proteins. Cupin_1and HSP70 in white-fleshed pitaya seeds, and cupin_1, HSP70, and HSP sti1-like in red-fleshed pitaya seeds were predicted to be allergens. The protein families that predicted allergens in pitaya seeds were similar to known allergens, indicating that these homologous allergens from different species may be cross-reactive.

Surprisingly, similar homologous allergens were exposed mostly by the airway but not the gastrointestinal tract, which indicates that the influence of natural environmental factors, especially pollen and dust mites, on food allergy is important. Growing lines of evidence have shown that non-oral routes of environmental exposure to food allergen homologs in pollen or house dust contribute to food sensitization and allergies. It has been reported that most patients who are allergic to fruits present pollen–food allergy syndrome (PFAS) because pollen-specific IgE can recognize homologous allergens in fruits that share the same epitopes [32]. Similarly, cross-reactivity has also been reported between dust mites and food allergens [33]. Therefore, considering the low frequency of patients eating fleshed pitaya, we assumed that those with allergies to pitaya were sensitized by pollen or dust mite inhalation in the air.

## 5. Conclusions

In this study, based on the clinical cases of pitaya fruit allergy, some potential allergens in seeds of *S. undatus* and *S. costaricensis* were identified using bottom-up proteomic comminated with immunoinformatics tools. The LC–MS/MS proteomics was used to investigate pitaya fruit seed protein bands at 20 kDa in SDS–PAGE gels because these bands contained the most reactive proteins toward IgE in the sera of allergic patients. The LC–MS/MS analysis with immunoinformatics prediction tools (Allermatch^TM^, Algpred 2.0, and AllerCatPro) attributed the most reactive proteins with specific IgE to cupin_1and HSP70 in white-fleshed pitaya seeds, and cupin_1, HSP70 and HSP sti1-like in red-fleshed pitaya seeds. Moreover, the expression of two and three potential allergens in white- and red-fleshed pitaya seeds, respectively, was further confirmed at the transcription level, and the CDSs for potential allergens were obtained from the *S. undatus* genome. To the best of our knowledge, this is the first report on allergens in white- and red-fleshed pitaya seeds. Compared with previous reports on allergies to pitaya fruit, a specific IgE for profilin (about 15 kDa) and LTP (about 10 kDa) was not found in the sera obtained from our clinical cases. This study provides valuable data toward a better understanding of the problem of pitaya allergy and can promote the identification of allergens from white- and red-fleshed pitaya seeds.

## Figures and Tables

**Figure 1 nutrients-14-01962-f001:**
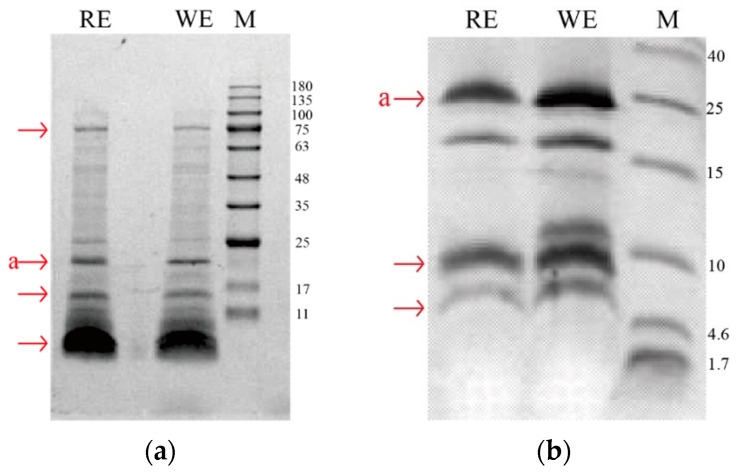
Separation on 4~20% Tris–Gly–SDS and 16.5% Tris–tricine polyacrylamide gels of white- and red-fleshed pitaya seeds protein extract. Lanes WE and RE correspond to white- and red-fleshed pitaya seeds protein extraction, respectively. Lane M: corresponds to protein markers and their corresponding molecular masses (kDa). Major protein bands are marked by red arrows: (**a**) pitaya seeds protein composition analysis by 4~20% Tris–Gly–SDS polyacrylamide gel and (**b**) pitaya seeds protein composition analysis by 16.5% Tris–tricine polyacrylamide gel.

**Figure 2 nutrients-14-01962-f002:**
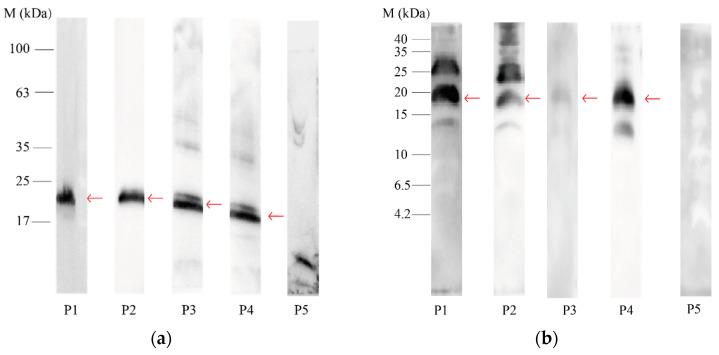
Immunoblot analysis of the serum IgE from five patients with allergy to pitaya (P1~P5) binding to white-fleshed pitaya seeds protein extraction. Approximately 20 kDa of IgE protein bands showing reactivity are marked with red arrows: (**a**) immunoblot analysis after protein separation of white-fleshed pitaya seed extracts on 4~20% Tris–Gly–SDS polyacrylamide gels; (**b**) immunoblot analysis after protein separation of white-fleshed pitaya seed extracts on 16.5% Tris–tricine polyacrylamide gels.

**Figure 3 nutrients-14-01962-f003:**
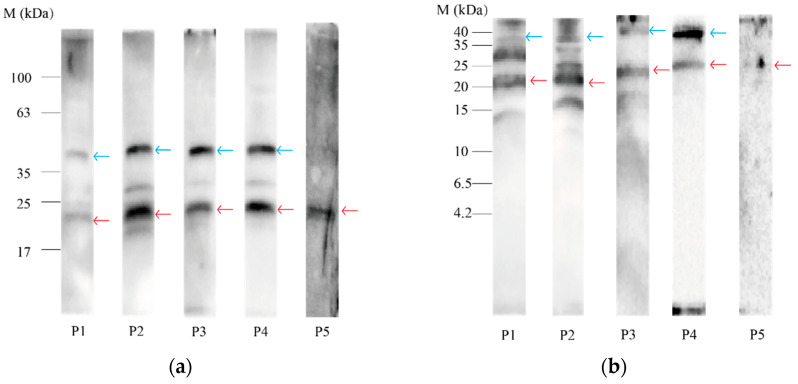
Immunoblot analysis of the serum IgE from five patients with allergy to pitaya (P1~P5) binding to red-fleshed pitaya seeds protein extraction. Approximately 20 kDa of IgE protein bands showing reactivity are marked with red arrows, approximately 40 kDa of IgE protein bands showing reactivity are marked with blue arrows: (**a**) immunoblot analysis after protein separation of red-fleshed pitaya seed extracts on 4~20% Tris–Gly–SDS polyacrylamide gels; (**b**) immunoblot analysis after protein separation of red-fleshed pitaya seed extracts on 16.5% Tris–tricine polyacrylamide gels.

**Figure 4 nutrients-14-01962-f004:**
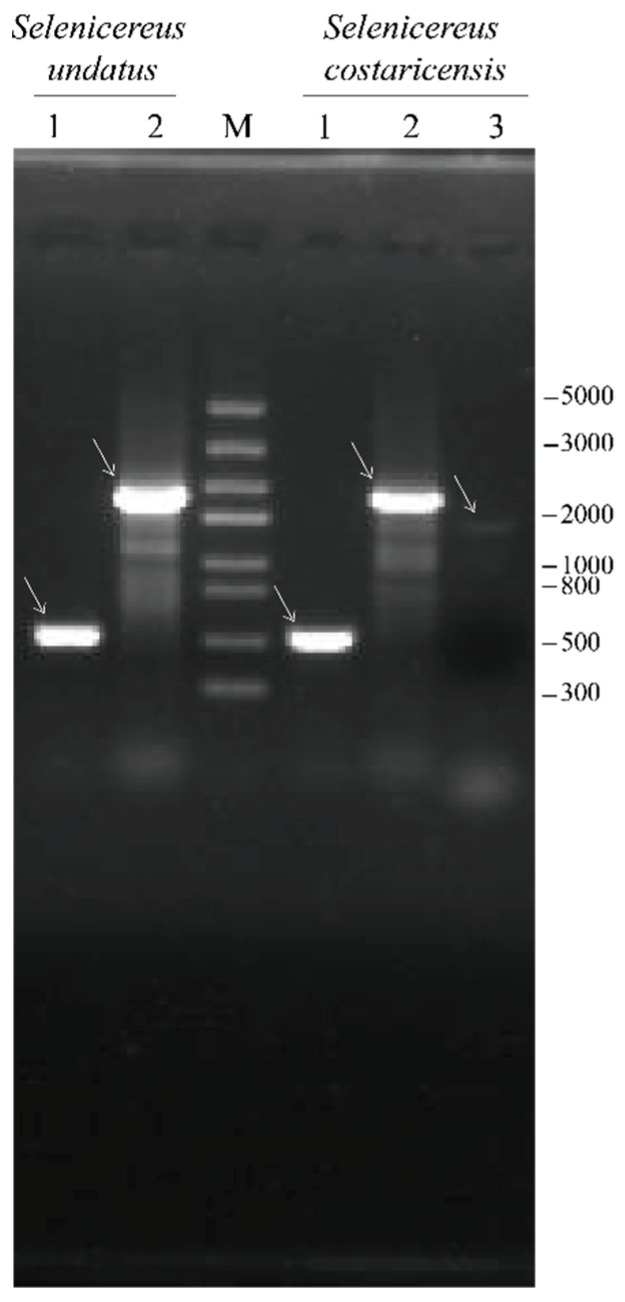
Agarose gel (1%) electrophoresis analysis of the RT-PCR products. Lane of 1, 2, and 3: correspond to the RT-PCR products for the CDSs of *cupin*, *hsp 70*, and *hsp sti 1*, respectively. The M line corresponds to DNA markers and their corresponding weight (bp). The main RT-PCR products are marked with white arrows.

**Table 1 nutrients-14-01962-t001:** Top 10% of proteins identified by LC–MS/MS in excised gel spots from white-fleshed pitaya seeds.

Protein Bands	N.	Description	Accession in UniProtKB Database	Score	Mass (Da)	Peptides (Unique)	Coverage (%)	emPAI
a	1	Uncharacterized protein (Fragment) OS = Beta vulgaris subsp. vulgaris	A0A0J8DSR1	158	26,488	6(6)	24.1	1.29
	2	Cupin type-1 domain-containing protein OS = Opuntia streptacantha	A0A7C8ZNT3	143	18,045	2(1)	12.7	0.41
	3	Uncharacterized protein OS = Spinacia oleracea	A0A0K9RTZ6	137	65,882	2(1)	4.7	0.1
	4	Uncharacterized protein OS = Chenopodium quinoa	A0A803LTG6	132	19,434	4(4)	27.5	1.22
	5	Heat shock protein 70 OS = Spinacia oleracea	A0A1I9TK81	106	71,808	3(3)	6.1	0.14

OS: organism species.

**Table 2 nutrients-14-01962-t002:** Top 10% of protein identified by LC–MS/MS in excised gel spot from red-fleshed pitaya seeds.

Protein Bands	N.	Description	Accession in UniProtKB Database	Score	Mass (Da)	Peptides (Unique)	Coverage (%)	emPAI
a	1	Proteasome subunit beta (Fragment) OS = Spinacia oleracea	A0A0K9S2G1	315	29,745	4(1)	20.4	1.1
	2	Uncharacterized protein (Fragment) OS = Opuntia streptacantha	A0A7C8YRP5	283	15,393	4(1)	37.8	3.02
	3	Cupin type-1 domain-containing protein OS = Opuntia streptacantha	A0A7C8ZNT3	274	18,045	2(1)	12.7	0.41
	4	Uncharacterized protein OS = Opuntia streptacantha	A0A7C9B091	248	64,965	5(5)	11	0.34
	5	Heat shock protein 70 OS = Spinacia oleracea	A0A1I9TK81	230	71,808	6(6)	11	0.43
	6	Actin 11 OS = Sesuvium portulacastrum	A0A1L5JKA9	206	41,929	6(6)	19.2	0.7
	7	Ribosomal_L18e/L15P domain-containing protein OS = Beta vulgaris subsp. Vulgaris	A0A0J8FEI6	182	20,923	4(3)	26.2	1.45
	8	Formamidase OS = Opuntia streptacantha	A0A7C9EWT0	182	52,325	3(3)	8.7	0.36
	9	Uncharacterized protein OS = Spinacia oleracea	A0A0K9RTZ6	179	65,882	2(1)	4.7	0.1
	10	Uncharacterized protein (Fragment) OS = Opuntia streptacantha	A0A7C8YRL3	170	15,770	4(2)	34.2	2.91
	11	Proteasome subunit beta OS = Opuntia streptacantha	A0A7C9D7H1	164	22,828	4(4)	15.6	1.27
	12	Cupin_5 domain-containing protein OS = Opuntia streptacantha	A0A7C9CU14	156	22,754	1(1)	6.5	0.32
	13	Uncharacterized protein OS = Opuntia streptacantha	A0A7C8ZZS4	146	22,123	4(1)	28	0.76
	14	GTP-binding nuclear protein OS = Spinacia oleracea	A0A0K9R796	131	25,564	4(3)	20.8	0.64
	15	Uncharacterized protein OS = Chenopodium quinoa	A0A803LTG6	129	19,434	3(0)	19.9	1.22
	16	Thioredoxin-dependent peroxiredoxin OS = Tamarix hispida	I0CC94	123	30,019	2(1)	10.9	0.52
	17	Uncharacterized protein OS = Opuntia streptacantha	A0A7C8Z5R6	122	22,142	4(1)	22.8	0.76

OS: organism species.

**Table 3 nutrients-14-01962-t003:** Potential allergens from white-fleshed pitaya seeds predicted using three online platforms for protein allergenicity.

Description of Potential Allergen (Accession in UniProtKB Database)	Protein Family for Potential Allergen	Predicted Most Similar Allergen	Protein Family for Predicted Most Similar Allergen	Predicted Most Similar Allergen Source	Predicted Most Similar Allergen Exposed Way
Cupin type-1 domain-containing protein OS = Opuntia streptacantha (A0A7C8ZNT3)	Cupin_1	Pollen allergen Coc n 1	Vicilin-like protein	*Cocos nucifera* (Coconut)	Airway
Uncharacterized protein OS = Spinacia oleracea (A0A0K9RTZ6)	Cupin_1	Pollen allergen Coc n 1	Vicilin-like protein	*Cocos nucifera* (Coconut)	Airway
Heat shock protein 70 OS = Spinacia oleracea (A0A1I9TK81)	Heat shock protein 70	Tyr p 28	Heat shock protein	*Tyrophagus putrescentiae* (Storage mite)	Airway

**Table 4 nutrients-14-01962-t004:** Potential allergens from red-fleshed pitaya seeds predicted using three online platforms for protein allergenicity.

Description of Potential Allergen (Accession in UniProtKB Database)	Protein Family for Potential Allergen	Predicted Most Similar Allergen	Protein Family for Predicted Most Similar Allergen	Predicted Most Similar Allergen Source	Predicted Most Similar Allergen Exposed Way
Cupin type-1 domain-containing protein OS = Opuntia streptacantha (A0A7C8ZNT3)	Cupin_1	Pollen allergen Coc n 1	vicilin-like protein	*Cocos nucifera* (Coconut)	Airway
Uncharacterized protein OS = Opuntia streptacantha (A0A7C9B091)	Heat shock protein Sti1-like	Hev b 5	Acidic protein	*Hevea brasiliensis* (Para rubber tree (latex))	Contact
Heat shock protein 70 OS = Spinacia oleracea (A0A1I9TK81)	Heat shock protein 70	Tyr p 28	Heat shock protein	*Tyrophagus putrescentiae* (Storage mite)	airway
Uncharacterized protein OS = Spinacia oleracea (A0A0K9RTZ6)	Cupin_1	Gly m Bd 28K	Cupin_1	*Glycine max* (Soybean)	Food
Thioredoxin-dependent peroxiredoxin OS = Tamarix hispida (I0CC94)	AhpC-TSA domain, 1-cysPrx_C domain	thiol peroxiredoxin	Peroxiredoxin, AhpC-type	*Bombyx mori* (domestic silkworm)	Airway

**Table 5 nutrients-14-01962-t005:** mRNA sequences of potential allergens Blast against *Selenicereus undatus* genome in the NCBI database.

Predicted Allergen (Accession in UniProtKB Database)	Accession of mRNA Sequences for Predicted Allergen in ENA Database	BLAST Search against *Selenicereus undatus* Genome in NCBI				Homology among Theoretical and Experimental Proteins
		E Value	Per. Ident	Acc. Len	Accession	
Cupin type-1 domain-containing protein OS = Opuntia streptacantha (A0A7C8ZNT3)	AFH74407	3.00 × 10^−158^	86.93	117,650,587	JACYFF010031955.1	85.03%
Heat shock protein 70 OS = Spinacia oleracea (A0A1I9TK81)	AOZ81415	0	80.61	112,022,934	JACYFF010031958.1	92.01%
Uncharacterized protein OS = Opuntia streptacantha (A0A7C9B091)	MBA4679171	0	87	109,661,751	JACYFF010031956.1	86.20%
Uncharacterized protein OS = Spinacia oleracea (A0A0K9RTZ6)	KNA22928	No significant similarity found.				
Thioredoxin-dependent peroxiredoxin OS = Tamarix hispida (I0CC94)	AFH74407	No significant similarity found.				

Homology among the theoretical and experimental proteins was determined by an alignment of the amino acids simulation translated from PCR products with the predicted allergens based on BLAST search of NCBI database.

## Data Availability

Not applicable.

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
