# Peer review of "Identification of Allergens in White- and Red-Fleshed Pitaya (*Selenicereus undatus* and *Selenicereus costaricensis*) Seeds Using Bottom-Up Proteomics Coupled with Immunoinformatics"

_nutrients, 2022, doi:10.3390/nu14091962_

Round 1

Reviewer 1 Report

The authors identify (white and red) pitaya seed allergens using five sera from volunteers reported to have symptoms after ingesting pitaya and standard molecular biological methods.  They identify a single dominant IgE-reactive band in red and white pitaya seed extract by immunoblot, migrating at approximately 20 kDa on tricine gels, using five pitaya allergic volunteer sera.  There was also a ~40kDa IgE-reactive band in red pitaya seed extract. While the methods and scientific approach presented are sound, several improvements must be made to this manuscript.  

The manuscript has numerous grammatical error that must be corrected.  There are many instances of incorrect or awkward grammar (too numerous to point out). 

The gel in Figure 1b appears to be upside down, is it oriented correctly?

It’s not clear what the blot in Figure 2b adds to the paper, consider removing it because it is not a good image and is difficult to interpret

Lines 214-215, this is difficult to interpret, there shouldn’t be so many different proteins identified from a 1-D gel slice, it suggests the proteins in the extracts were highly degraded and makes the interpretation of the LC-MS/MS data more difficult 

The title of Table 1 and 2 are identical, this appears to be a mistake, and Table 2 should be ‘…red-fleshed pitata…’

Could you add a column indicating the percentage that each protein represents in Tables 1 and 2? 

Lines 230-254 and Table 3 and 4, Coc n 1 has a predicted mass of 53 kDa and Tyr p  (a mite protein) predicted mass of 76 kDa, so this also suggests that the extracts could have a large amount of degraded protein in them making it difficult to interpret the LC-MS/MS data

To allow a solid conclusion to be made about which proteins in pitaya seeds are bound by the allergic volunteer IgE the authors need to make recombinant versions of each of the proposed pitaya seed allergens and determine which, if any, of the recombinant proteins the pitaya allergic volunteer sera-IgE bind

Author Response

Dear editor and reviewer:

Thank you for your letter and the reviewer’s comments on our manuscript entitled “Identification of Allergens in White-fleshed and Red-fleshed Pitaya (Selenicereus undatus and Selenicereus costaricensis) Seeds by Bottom-Up Proteomics Coupled with Immunoinfor-matics” (No. nutrients-1687933). These comments are very helpful for revising and improving our manuscript, as well as the important guiding significance to our research. We have studied the comments carefully and made corrections and adjuestion which we hope meet with approval. The main revisions has been marked up using the “Track” in the manuscript and the responds to the reviewers’ comments are as shown on respond document.

Reviewer 2 Report

Minor

Line 15 rewrite and correct: On the protein spot about 20 kDa displayed a clear reaction with serum IgE. And the protein spot was excised, in-gel digested, and analyzed by liquid chromatography-tandemmass spectrometry

Line 36: why as hidden allergens?

Line 49: Rewrite “However, whether there are allergens hidden in white-fleshed pitaya and pitaya 49 seeds are for further study”

Line 119: Correct as De Angelis et al. not by Elisabetta, et al.[14].

Line 153: correct the significative digits in 6.9±0.098 mg/mL and 6.8±0.017 mg/mL extracted

Line 190: Rewrite “As appearing from figures, about the 20 kDa protein band from both white-fleshed 190 and red-fleshed pitaya had reactivity to IgE, and the bands were identified by most se- 191 rum from patients.”

Line 335: Please consider and compare with the paper in Talanta, 2022;240:123188 to explain this concept: So, in order to narrow the range of potential allergens, some allergen online prediction platforms were used including Allermatchtm, Algpred 2.0 and  AllerCatPro .The development of Allermatchtm was based on Codex Alimentarius Commission (2003) Guideline for the Conduct of Food Safety Assessment of Foods derived from Recombinant-DNA Plants CAC/GL 45-2003 [32]. Be careful since authors are excluding many proteins for allergen match.

Major points:

The English language and style need major correction, it’s too weak and full of errors (grammatical and spelling).

Which differences rely upon Tables 1 and 2? How is possible to have so many results from one spot at 20 kDa? The authors found a lot of proteins in one spot also being of very high MW. Please explain these results.

The major concerns are about the presentation of MS/MS data. Please insert in SI also tables with identified peptides and protein coverages. Please insert also the % of homology found among theoretical and experimental proteins.

Author Response

(The authors gave the same response as above.)

Round 2

Reviewer 1 Report

The manuscript still has numerous grammatical errors that will make it difficult for readers to understand.  That said, the manuscript has been improved enough that the general message and important points can be reliably understood.   

Author Response

Dear editor and reviewer:

Thank you for your letter and the reviewer’s comments on our manuscript entitled “Identification of Allergens in White-fleshed and Red-fleshed Pitaya (Selenicereus undatus and Selenicereus costaricensis) Seeds by Bottom-Up Proteomics Coupled with Immunoinfor-matics” (No. nutrients-1687933). These comments are very helpful for revising and improving our manuscript, as well as the important guiding significance to our research. We have studied the comments carefully and made corrections and adjuestion which we hope meet with approval. The main revisions has been marked up using the “Track” in the manuscript and the responds to the reviewers’ comments are shown as attachment.

Reviewer 2 Report

6.89±0.0976 mg/mL and 6.80±0.0170 correct as 6.89±0.01 mg/mL and 6.80±0.02

The authors should be careful since grammar errors are recurrent. Please, check.

Author Response

(The authors gave the same response as above.)
